# Strategy to Reduce the Collective Equivalent Dose for the Lens of the Physician’s Eye Using Short Radiation Protection Curtains to Prevent Cataracts

**DOI:** 10.3390/diagnostics11081415

**Published:** 2021-08-05

**Authors:** Koichi Nakagami, Takashi Moritake, Keisuke Nagamoto, Koichi Morota, Satoru Matsuzaki, Tomoko Kuriyama, Naoki Kunugita

**Affiliations:** 1Department of Radiology, Hospital of the University of Occupational and Environmental Health, 1-1 Iseigaoka, Yahatanishi-ku, Kitakyushu, Fukuoka 807-8556, Japan; nakagami@clnc.uoeh-u.ac.jp (K.N.); k-nagamoto0201@clnc.uoeh-u.ac.jp (K.N.); 2Department of Occupational and Community Health Nursing, School of Health Sciences, University of Occupational and Environmental Health, 1-1 Iseigaoka, Yahatanishi-ku, Kitakyushu, Fukuoka 807-8555, Japan; qqey494d@adagio.ocn.ne.jp (S.M.); kuritomo@med.uoeh-u.ac.jp (T.K.); kunugita@med.uoeh-u.ac.jp (N.K.); 3Department of Radiation Regulatory Research Group, National Institute of Radiological Sciences, Quantum Life and Medical Science Directorate, National Institute for Quantum and Radiological Science and Technology, 4-9-1 Anagawa, Inage-ku, Chiba 263-8555, Japan; 4Department of Radiology, Shinkomonji Hospital, 2-5 Dairishinmachi, Moji-ku, Kitakyushu, Fukuoka 800-0057, Japan; k_morota23@yahoo.co.jp

**Keywords:** radiation protective curtain, fluorography, lens dose, collective dose, cataract, population strategy, high-risk strategy

## Abstract

A short curtain that improves on the low versatility of existing long curtains was developed as a dedicated radiation protective device for the over-table tube fluorographic imaging units. The effect of this short curtain in preventing cataracts was then examined. First, the physician lens dose reduction rate was obtained at the position of the lens. Next, the reduction rate in the collective equivalent dose for the lens of the physician’s eye was estimated. The results showed that lens dose reduction rates with the long curtain and the short curtain were 88.9% (literature-based value) and 17.6%, respectively, higher with the long curtain. In our hospital, the reduction rate in the collective equivalent dose for the lens of the physician’s eye was 9.8% and 17.6% with a procedures mixture, using the long curtain where technically possible and no curtain in all other procedures, and the short curtain in all procedures, respectively, higher with the short curtain. Moreover, a best available for curtains raised the reduction rate in the collective equivalent dose for the lens of the physician’s eye a maximum of 25.5%. By introducing the short curtain, it can be expected to have an effect in preventing cataracts in medical staff.

## 1. Introduction

The International Commission on Radiation Protection (ICRP), in its 2011 “ICRP Statement on Tissue Reactions,” recommended “an equivalent dose limit for the lens of the eye of 20 mSv year, averaged over defined periods of 5 years, with no single year exceeding 50 mSv” [1]. Considering this, reducing the lens exposure of medical personnel is an urgent issue from the perspective of cataract prevention.

According to ICRP Pub.117, the fluoroscopy examination and procedures performed by gastroenterologists are complex, and fluoroscopy time can reach as high as 93 min, making physician exposure a problem [2]. The equivalent dose for the lens of the eye of medical personnel who perform endoscopic retrograde cholangiopancreatography (ERCP) is reported to be 15.5–210 μSv per patient [3,4,5,6,7,8]. In addition, it has been reported that, when gastroenterologists perform ERCP without any lens protection, the equivalent dose limit for the lens of the eye (20 mSv/year) is exceeded [9,10]. Therefore, the guidelines of the World Gastroenterology Organization recommend the use of lead glasses or facemasks or visors of lower lead equivalence that cover the whole of the face [11]. The International Atomic Energy Agency (IAEA) has released information showing that exposure can be reduced with the use of lead glasses, ceiling-suspended lead acrylic shields, or movable shields, and teaches the importance of lens protection [12]. However, a 2019 survey of endoscopists in the United States found that the percentage who responded that they always or almost always used lead glasses was 30%, whereas the rate of use of ceiling-suspended lead acrylic shields or movable shields was 10%. Thus, it would be difficult to conclude that physicians adopt sufficient lens protection [13].

Fluoroscopy systems used for performing interventional procedures in radiology, neuroradiology and cardiology are generally configured with the x-ray tube positioned beneath the patient table to decrease the amount of backscatter radiations to a physician [14]. On the other hand, for certain procedures such as upper gastrointestinal examinations, to make a sufficient space above the patient’s body so that the physician can directly see and touch the patient, over-table fluorographic imaging units are used. In the latter case, the exposure of the physician comes to be large, so it is necessary to take standard radiation protection practices concerning time, distance, and shielding [14]. In Japan, radiation protective curtains, hereinafter called long curtains, have been developed and are widely used as protective devices that can be mounted on over-table tube fluorographic imaging units [15,16]. Long curtains consist of four leaded sheets, and they are attached to the X-ray tube cover. By completely enveloping the X-ray tube and the direct X-ray beams that are irradiating the patient, the physician’s exposure to scattered radiation can be reduced by about 90% [16]. However, long curtains are not transparent, so the physician cannot observe the patient. It has been pointed out that, because of this, the physician cannot perform procedures that involve lightly tapping or pressing the skin of the patient directly above the affected area from the outside, nor can the physician insert or extract a percutaneous drain while viewing the fluoroscopic image [15]. Another problem is that, when the table is tilted, the weight of the curtain makes it hang down so that it impedes the X-ray beams, greatly constraining the procedure.

To solve these problems, in this study, a short, leaded radiation protective curtain that allows the patient to be observed and does not enter the fluoroscopic visual field even when the table is tilted was developed. Throughout this paper, this is referred to as a short curtain. The radiation protection effect of this short curtain was also compared with that of long curtains from the two perspectives of the physician lens dose reduction rate and the reduction rate in the collective equivalent dose for the lens of the eye of all physicians involved in fluoroscopy examinations in the hospital.

## 2. Materials and Methods

### 2.1. Development of the Short Curtain

The short curtain consists of a 100 cm × 30 cm × 0.5 cm leaded curtain section and a section that connects to the X-ray tube collimator (Figure 1a). The curtain surrounds the X-ray tube collimator (Figure 1b). This can reduce the scattered radiation that is produced mainly from the X-ray tube collimator. The short curtain weighs 1.6 kg and has a 0.2-cm-thick acrylic plate sandwiched together with the lead sheet (equivalent to 0.13-mm lead thickness) by urethane sheet surface material (Figure 1c) so that the shape does not change from its own weight when the table is tilted (Figure 1d). A hook-and-loop fastener was sewn onto the top of the short curtain. The other fastener was attached to the X-ray tube collimator by powerful double-sided tape made especially for plastic.

### 2.2. Analysis of the Air Dose Rate

As shown in Figure 2, a 50 cm × 60 cm grid with 10 cm spacing was established near the head of the physician, and the air dose at each lattice point in the grid was measured with a radio-photoluminescence glass dosimeter (GD-302M; Chiyoda Technol Corporation, Tokyo, Japan). X-ray fluoroscopy was performed using an over-table tube fluorographic imaging unit (Sonialvision G4; Shimadzu Corporation, Kyoto, Japan). A water body phantom (Iken Engineering Co.,Ltd, Tokyo, Japan) was placed on the table as a source of scattered radiation, and this was followed by X-ray exposure for 10 min with tube voltage of 85 kVp, tube current of 2.3 mA, source to image receptor distance of 120 cm, and a square irradiation field of 30 cm on one side. The air dose rate in the 10-cm squares of the grid was obtained as the average of the air dose rate at the lattice points in the 4 corners of each square.

### 2.3. Analysis of the Physician’s Lens Presence Rate in Grid Squares

Main four fluoroscopic procedures were analyzed in 20 patients at our hospital from 1 January 2017, to the end of May 2021 (myelography in 5 cases, nerve root block in 5 cases, endoscopic retrograde cholangiopancreatography in 5 cases, upper gastrointestinal series in 5 cases). The physician’s actions were observed visually by the radiological technologist who charged in each case. Observation was performed with an interval of fifteen seconds during procedure. We counted how many times the physician’s eyes were in which grid square. The lens presence rate was determined according to the following equation:PR (row, column) = Ct (row, column)/Total count,(1)
where PR (row, column) is the lens presence rate at the grid square of Area (row, column), Ct (row, column) is the number of times the physician’s eyes are in Area (row, column), and Total count is the number of times counted throughout procedure.

### 2.4. Analysis of the Physician Lens Dose Reduction Rate with the Short Curtain

The air dose rate of each Area (row, column) when using and not using the short curtain was obtained, and the crystalline lens dose reduction rate while using the short curtain (DRR_Short_) was determined according to the following equation:DRR_Short_ (row, column) = (A (row, column) − B (row, column))/A (row, column),(2)
where A (row, column) is the air dose rate at Area (row, column) without the short curtain, and B (row, column) is that at Area (row, column) with the short curtain.

The total lens dose reduction rate of the physician with the short curtain (Total DRR_Short_) was derived as follows:Total DRR_Short_ = Σ PR (row, column) × DRR_Short_ (row, column).(3)

### 2.5. Analysis of the Physician Lens Dose Reduction Rate with the Long Curtain

The total lens dose reduction rate with the long curtain (Total DRR_Long_) was previously reported to be 88.9% [15,16]. In the present study, Total DRR_Long_ = 0.889 was applied to four fluoroscopic procedures: ERCP, endoscopic injection sclerotherapy (EIS), endoscopic sphincter papillotomy (EST), and endoscopic retrograde biliary drainage (ERBD).

### 2.6. Analysis of the Reduction Rate in the Collective Equivalent Dose for the Lens of the Physician’s Eye with the Short Curtain and Long Curtain

Fluoroscopy examinations using an over-table tube fluorographic imaging unit that were performed between January 1 and December 31, 2017 at our hospital were extracted from our radiation information system (RIS), and the air kerma-area product (P_KA_) for each test procedure was totaled (Collective P_KA_). In the present study, fluoroscopic examinations performed in an operating room for surgery and barium swallow tests were excluded, since the positional relationship between the physician and the X-ray tube differs significantly from that in other procedures.

It was first assumed that the P_KA_ value is proportional to the equivalent dose for the lens of the physician’s eye (H_Lens_), and coefficient *t* is defined as follows to convert from the P_KA_ value to the equivalent dose for the lens of the physician’s eye (H_Lens_) when a curtain is not used.
*t* = H_Lens_/P_KA_ (Sv·Gy^−1^·m^−2^).(4)

Using this coefficient *t*, the collective equivalent dose for the lens of the physician’s eye (Collective H_Lens_) when a curtain is not used was estimated as follows:Collective H_Lens_ = Collective P_KA_ × *t.*(5)
next, by multiplying the Collective H_Lens_ by the Total DRR with the long curtain and short curtain, the Collective H_Lens_ of the physician was calculated for three situations: when the long curtain was used; when the short curtain was used; and when the best curtain available (long curtain or short curtain) was used. The reduction rate in the collective equivalent dose for the lens of the physician’s eyes (Collective H_Lens_ RR) in each situation was calculated according to the following formula:Collective H_Lens_ RR_Long/Short/Both_ = (Collective H_Lens, without curtain_ − Collective H_Lens, with Long/Short/Both curtain_)/Collective H_Lens, without curtain_,(6)
note, however, that Collective H_Lens, without curtain_ is the value when no curtain was used, and Collective H_Lens, with Long/Short/Both curtain_ is the value when the long curtain, short curtain, or both curtains were used.

## 3. Results

### 3.1. Physician Lens Dose Reduction Rate with the Short Curtain

Figure 3 shows the dose rate (µGy/min) distribution when the short curtain was (Figure 3a) and was not (Figure 3b) used. When the physician performs a procedure standing right beside the operating table, the lens is in Area (c, 5), Area (d, 5) or Area (e, 5) (Figure 4). The physician adopts a forward-leaning posture when removing or inserting a drain and looking at his or her hands, and the lens is in Area (d, 4), Area (e, 4) or Area (f, 4) (Figure 4). Taking a weighted average, the Total DRR_Short_ was 0.176 (17.6%) (Table 1).

### 3.2. Reduction Rate in the Collective Equivalent Dose for the Lens of the Physician’s Eye with the Short Curtain and the Long Curtain

The Collective H_Lens_ RR was higher with use of the short curtain in all procedures (0.176, 17.6%) than that with use of the long curtain where technically possible and no curtain in all other procedures (0.098, 9.8%), though the shielding effect was higher for the long curtain (0.889, 88.9%) than that for the short curtain (0.176, 17.6%).

When either the long curtain or short curtain was used reliably in all cases, the Collective H_Lens_ RR reached a maximum of 0.255 (25.5%) (Table 2).

## 4. Discussion

Ten years have passed since the ICRP recommendations of 2011, and, still today, measures to protect physicians against lens exposure cannot be considered sufficient [13,17,18]. The use of radiation protective devices is necessary to reduce the lens dose in procedures that use a fluorographic imaging unit [19]. Various studies of the dose reduction rate with the protective devices used in fluoroscopy examinations have been performed [5,10,15,16,20,21,22], but the present study is the first to verify the exposure reduction rate with radiation protective devices from the perspective of collective equivalent dose for the lens of the eye in the entire hospital.

Although the lens dose reduction rate was much higher with the long curtain than with the short curtain (Total DRR_Long_: 0.889, Total DRR_Short_: 0.176), viewed from the perspective of collective equivalent dose for the lens of the physician’s eye, the reduction rate was calculated to be lower for a procedures mixture, using the long curtain where technically possible and no curtain in all other procedures (Collective H_Lens_ RR_Long_: 0.098) versus using the short curtain in all procedures (Collective H_Lens_ RR_Short_: 0.176) (Table 2). Physicians often need to tap or press the patient’s body, to remove or insert the percutaneous drainage tubes, or to tilt the patient table during procedures. In these cases, the long curtain is not available. Therefore, the use of the long curtain is limited in endoscopy, such as ERCP, EIS, EST, and ERBD, which the physicians do not need to see or touch the patient’s body or to tilt the patient table. This is thought to be one reason why long curtains are used in only 15% of fluoroscopy examinations in our hospital ((188 + 37 + 32 + 17)/1834 = 0.149 Table 2). Even though the dose reduction rate is low with short curtains alone, the results suggest that, when they are widely used, the reduction in the collective equivalent dose for the lens of the physician’s eye is superior to that with long curtains.

The population strategy and the high-risk strategy proposed by Rose in 1985 [23] are strategic intervention methods to promote health and disease prevention in given populations, and they are important concepts in terms of public health in preventive medicine [24,25,26,27]. Radiation cataracts are classified as a tissue reaction (deterministic effect), and selection of a high-risk strategy is appropriate. That is, for fluoroscopic techniques such as ERCP in which exposure levels are high, implementation of reliable prevention measures with a long curtain is effective. However, the threshold dose for radiation cataracts remains controversial [28,29], and further research is needed [30]. The possibility that there is a stochastic effect with no threshold dose for radiation cataracts has also been suggested [1]. Therefore, interventions with the population strategy, that is, preventive measures with the short curtain, which has high versatility even if the protective performance is imperfect, is thought to be effective.

The limitations of this study are that P_KA_ was assumed to be proportional to H_Lens_, and the conversion coefficient from P_KA_ to H_Lens_ was fixed as *t*. Therefore, if the physician’s standing position, the fluorographic imaging unity, the fluorography conditions set, other protective devices simultaneously used, or other factors differ, there is a possibility that *t* will change significantly with each case, and the collective equivalent dose for the lens of the physician’s eye cannot be correctly evaluated [31,32,33]. In addition, at medical institutions where a high percentage of fluoroscopic techniques can be done with the use of a long curtain, there is a possibility that not much reduction can be expected in the collective equivalent dose for the lens of the physician’s eye with short curtains.

## 5. Conclusions

Compared with the physician’s lens dose reduction rate with a long curtain of 0.889 (88.9%) reported in the literature, the lens dose reduction rate with the short curtain made in this study was found experimentally to be 0.176 (17.6%). Looking at fluoroscopy examinations performed in our hospitals overall, the reduction rate in the collective equivalent dose for the lens of the physician’s eye was 0.098 (9.8%) with a procedures mixture, using the long curtain where technically possible and no curtain in all other procedures, and 0.176 (17.6%) with the short curtain in all procedures, reversing the lens dose reduction effect. By always fitting equipment with either a long curtain or a short curtain, the reduction rate in the collective equivalent dose for the lens of the physician’s eye reaches a maximum of 0.255 (25.5%). With the introduction of highly versatile short curtains, the physician collective equivalent dose for the lens of the eye in fluoroscopy examinations in the entire hospital is reduced, and it was shown that a cataract prevention effect in medical personnel can be expected.

## Figures and Tables

**Figure 1 diagnostics-11-01415-f001:**
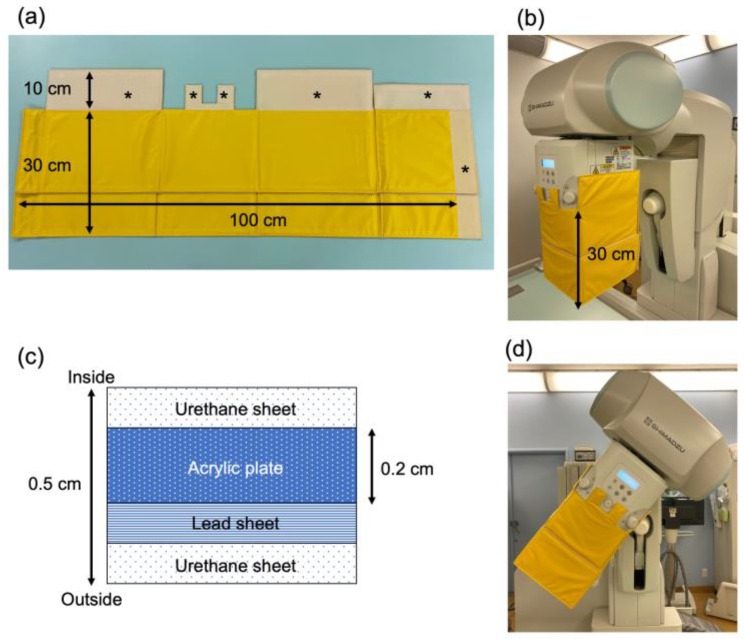
Composition and structure of the short curtain. (**a**) Short curtain composition. Asterisks show portions where hook and loop fasteners are sewn onto the short curtain. (**b**) Short curtain attached to the fluorographic imaging unit. (**c**) Cross-section of short curtain shielding material. (**d**) Short curtain when attached to fluorographic imaging unit and the unit is tilted.

**Figure 2 diagnostics-11-01415-f002:**
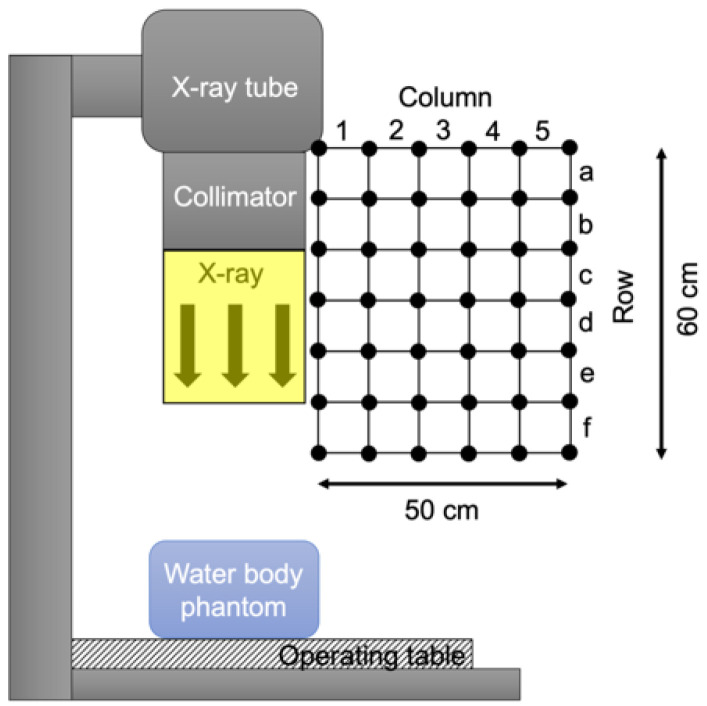
Setting for scattered radiation distribution measurements near the physician’s lens of the eye. The black circles show the placement of radio-photoluminescence glass dosimeters (RPLD). Areas (row, column) are set with a grid with 10 cm spacing.

**Figure 3 diagnostics-11-01415-f003:**
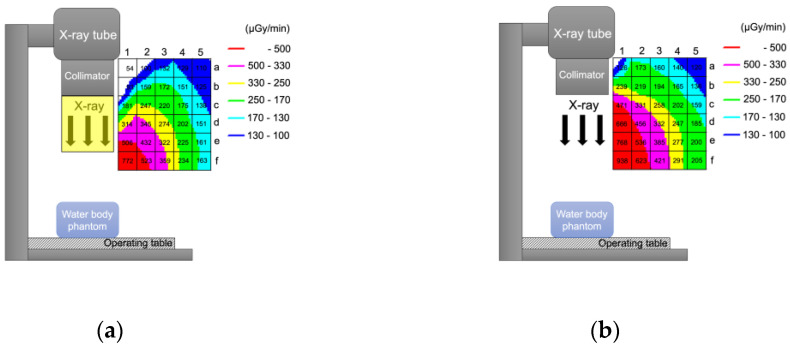
Change in scattered radiation distribution when a short curtain is used. (**a**) When a short curtain is used. (**b**) When a short curtain is not used. Numbers in the grid square represent the measured dose rate (μGy/min).

**Figure 4 diagnostics-11-01415-f004:**
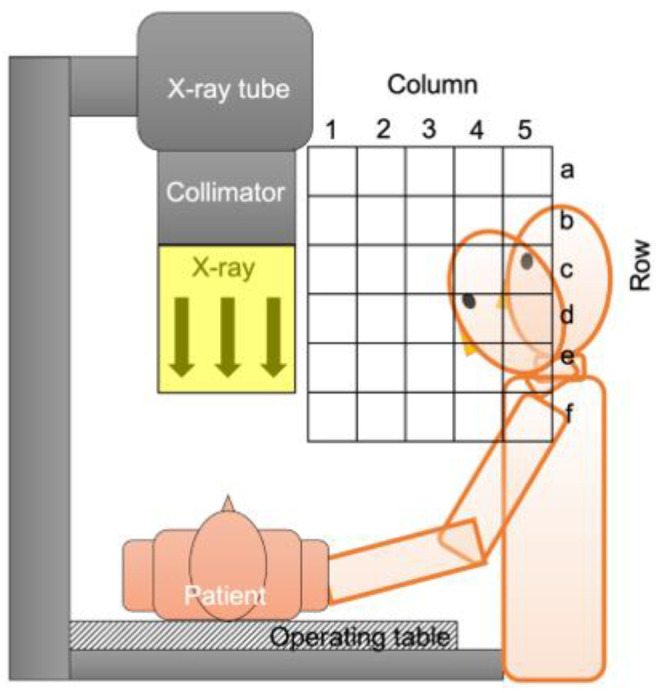
Analysis of physician’s lens position. The physician’s lens is mostly located in Area (d, 4), Area (e, 4) and Area (f, 4) when leaning forward to perform a task, and in Area (c, 5), Area (d, 5) and Area (e, 5) when standing upright to perform a task.

**Table 1 diagnostics-11-01415-t001:** Lens dose reduction rate with the short curtain.

Localization of the Operator’s Crystalline Lens, Area (Raw, Column)	Area (d, 4)	Area (e, 4)	Area (f, 4)	Area (c, 5)	Area (d, 5)	Area (e, 5)	Total Lens Dose Reduction Rate (Total DRR_Short_) ***
Lens presence rate, PR (raw, column) * (*n* = 20)	0.015 (1.5%)	0.049 (4.9%)	0.113 (11.3%)	0.183 (18.3%)	0.581 (58.1%)	0.058 (5.8%)	
Air dose rate without short curtain, A (raw, column) [μGy/min] (*n* = 3)	247	277	291	159	185	200	
Air dose rate with short curtain, B (raw, column) [μGy/min] (*n* = 3)	202	225	234	138	151	161	
Lens dose reduction rate, DRR_Short_ (raw, column) **	0.182 (18.2%)	0.188 (18.8%)	0.196 (19.6%)	0.132 (13.2%)	0.184 (18.4%)	0.195 (19.5%)	0.176 (17.6%)

* PR (raw, column) = The number of times located in the Area (raw, column) (Ct (raw, column))/The total number of times counted (Total count). ** DRR_Short_ (raw, column) = (A (raw, column) − B (raw, column))/A (raw, column), where A (row, column) is the air dose rate at Area (row, column) without the short curtain, and B (row, column) is that at Area (row, column) with the short curtain. *** Total DRR_Short_ = Σ PR (raw, column) × DRR_Short_ (raw, column).

**Table 2 diagnostics-11-01415-t002:** Reduction rate in the collective equivalent dose for the lens of the physician’s eye with the long curtain and the short curtain.

Fluorography Procedures Using Over-Table Tube Fluorographic Imaging Unit	Number of Cases *	Collective P_KA_ ** (Gy·m^2^)	Physician’s Collective H_Lens_						
			Without Curtain	With Long Curtain		With Short Curtain		With Best Curtain Available	
			Collective H_Lens_ (Sv)	Adopted Total DRR	Collective H_Lens_ (Sv)	Adopted Total DRR	Collective H_Lens_ (Sv)	Adopted Total DRR	Collective H_Lens_ (Sv)
Endoscopic retrograde cholangiopancreatography (ERCP)	188	0.483	0.483 *t*	0.889	0.054 *t*	0.176	0.398 *t*	0.889	0.054 *t*
Endoscopic injection sclerotherapy (EIS)	37	0.079	0.079 *t*	0.889	0.009 *t*	0.176	0.065 *t*	0.889	0.009 *t*
Endoscopic sphincterotomy (EST)	32	0.096	0.096 *t*	0.889	0.011 *t*	0.176	0.079 *t*	0.889	0.011 *t*
Endoscopic retrograde biliary drainage (ERBD)	17	0.033	0.033 *t*	0.889	0.004 *t*	0.176	0.027 *t*	0.889	0.004 *t*
Upper gastrointestinal series	171	0.776	0.776 *t*	N.A.	0.776 *t*	0.176	0.640 *t*	0.176	0.640 *t*
Barium enema	143	1.210	1.210 *t*	N.A.	1.210 *t*	0.176	0.997 *t*	0.176	0.997 *t*
Nerve root block	138	0.095	0.095 *t*	N.A.	0.095 *t*	0.176	0.078 *t*	0.176	0.078 *t*
Drain tube change	111	0.203	0.203 *t*	N.A.	0.203 *t*	0.176	0.167 *t*	0.176	0.167 *t*
Lumbar myelography	106	0.494	0.494 *t*	N.A.	0.494 *t*	0.176	0.407 *t*	0.176	0.407 *t*
Ileus tube follow-up	76	0.462	0.462 *t*	N.A.	0.462 *t*	0.176	0.381 *t*	0.176	0.381 *t*
Ileus tube insertion	71	0.807	0.807 *t*	N.A.	0.807 *t*	0.176	0.665 *t*	0.176	0.665 *t*
Percutaneous transhepatic cholangiodrainage	37	0.104	0.104 *t*	N.A.	0.104 *t*	0.176	0.086 *t*	0.176	0.086 *t*
Intravenous hyperalimentation catheter insertion	33	0.015	0.015 *t*	N.A.	0.015 *t*	0.176	0.013 *t*	0.176	0.013 *t*
Central venous catheter placement	32	0.030	0.030 *t*	N.A.	0.030 *t*	0.176	0.025 *t*	0.176	0.025 *t*
Percutaneous transhepatic gallbladder drainage	14	0.015	0.015 *t*	N.A.	0.015 *t*	0.176	0.012 *t*	0.176	0.012 *t*
Small bowel radiography	15	0.112	0.112 *t*	N.A.	0.112 *t*	0.176	0.092 *t*	0.176	0.092 *t*
Others	613	1.224	1.224 *t*	N.A.	1.224 *t*	0.176	1.009 *t*	0.176	1.009 *t*
Total	1834	6.239	6.239 *t*		5.625 *t*		5.141 *t*		4.649 *t*
Collective H_Lens_ Reduction Rate (Collective H_Lens_ RR)					0.098 (9.8%)		0.176 (17.6%)		0.255 (25.5%)

* Number of cases conducted in our hospital within one year (1 January to 31 December 2017). ** Sum of the P_KA_ values in our hospital within one year (1 January to 31 December 2017). P_KA_: The air kerma-area product. H_Lens_: The equivalent dose for the lens of the physician’s eye. Total DRR: The total lens dose reduction rate using a radiation protective curtain. *t*: The coefficient to convert from the P_KA_ value to the equivalent dose for the lens of the physician’s eye. N.A.: Not available: Protective device cannot be used because operative field cannot be seen, bed or X-ray tube cannot be tilted, or other reasons.

## Data Availability

No new data were created or analyzed in this study. Data sharing is not applicable to this article.

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
