# Peer review of "Strategy to Reduce the Collective Equivalent Dose for the Lens of the Physician’s Eye Using Short Radiation Protection Curtains to Prevent Cataracts"

_diagnostics, 2021, doi:10.3390/diagnostics11081415_

Round 1

Reviewer 1 Report

The manuscript, “Strategy to reduce the collective equivalent dose for the lens of the physician’s eye using short radiation protection curtains to prevent cataracts” by K. Nakagami, et al. introduced a short and leaded radiation protective curtain to reduce the collective equivalent dose for the lens of the physician’s eye for the over-table tube fluorographic imaging.  I would like to recommend this manuscript for publication in Diagnostics after revisions.  Comments and questions to authors should be addressed;

- The measurement took for 10 minutes.  Is there any special reason?

- The description for Figure 1c needs to be prior to the one for Figure 1d.

- It is better to label Figure 3a prior to Figure 3b.

- Is the legend in Figure 3 showed Max-Min of the dose rate?  Then it is better to display ‘ – 500’ instead of ‘500 – ’.

Reviewer 2 Report

General issues:
1) As an international reviewer I do not understand why over-table x-ray tubes are used for many procedures in Japan. In adult examinations below-table x-ray tubes should be used whenever possible, significantly reducing the exposure of the radiologist. Please discuss this aspect in your paper. You might want to reference this paper: doi: 10.2214/AJR.16.16454

2) Why did you only focus on the eye lens? There are other radiation sensitive organs in the scatter radiation field in your set-up, partially not being covered by the lead apron, e.g. the head/neck glandular tissue. Please address this issue.

3) What is the impact of radiation protection glasses? In europe these glasses are used by most interventional radiologist today. Please measure the air dose rate with and without the short curtain again, placing lead glass glasses between the x-ray source and the radio-photoluminescence glass dosimeters. This would significantly add to the value of your paper.

4) I would recommend native English language editing of the manuscript for better understandability.

Specific issues:
0) page 1, line 25:"The results showed that lens dose reduction rates with the long curtain and the short curtain were 24 0.889 (literature-based value) and 0.176, respectively"

Please give all values in the abstract as % value. This is easier to understand.
Same applies to the table 1.

1) page 4, line 111: "The physician’s actions were observed visually by the radiological technologist who charged in each case with an interval of fifteen seconds during procedure, and an analysis was done of the area (Area (row, column)) surrounded by the grid square in which the physician’s eyes were located, the number of times located in that area (Ct (row, column)), and the total number of times counted (Total count). The lens presence rate (PR (row, column)) was determined as the Ct (row, column) and the Total count ratio (Ct (row, column)/Total count) in each Area (row, column)."

This sentence is hard to understand. Please divide in multiple sentences and rephrase for better understandability.

2) page 4, line 136: "In the present study, operating room fluoro-136 scopic examinations that use a portable fluorographic imaging unit and fluoroscopic examination of swallowing and esophagography, in which the positional relationship between the physician and the X-ray tube differs significantly from that in other procedures, such as when the patient is put in a standing position, were excluded."

Please also use rephrase with shorter sentences.

3) page 5, line 149: "and when both the long 149 curtain and short curtain were used"

This desciption is misleading. To my understanding you calculated the lens exposition with the best curtain available for different examination. Is this correct? Then please rephrase accordingly.
Same applies to table 2, last column.

4) page 5, line 163: "the Total DRRShort was 0.176"

Please add the unit μGy/min.

5) page 5, line 164: Figure 3

The color-coding of the dose rate is nice. Could you please add the measured number of the dose rate to each square of the grid?

6) page 6, line 178: "The Collective HLens RR was higher (Collective HLens RRLong: 0.098; Collective HLens 178 RRShort: 0.176) with use of the short curtain, which has a low shielding effect (Total DRRShort: 0.176), in all cases than with use of the long curtain, which has a high shielding effect (Total DRRLong: 0.889) in ERCP, EIS, EST, and ERBD only."

Please rephrase for better understandability.

7) page 6, line 186: "measures to protect physicians against lens exposure cannot considered sufficient"

I assume you mean "cannot be considered sufficient".

8) page 6, line 195: "the reduction rate was conversely higher with the short curtain than with the long curtain (Collective HLens 196 RRLong: 0.098, Collective HLens RRShort: 0.176)"

This statement is overabbreviated and not correct. The reduction rate was calculated to be higher for a procedures mixture, using the long curtain where technically possible and no curtain in all others procedures versus using the short curtain in all procedures. Please rephrase.

9) page 6, line 197: "This is because, with the long curtain procedures in which the physician directly taps or presses the affected area from the skin, procedures in which percutaneous drainage tubes are removed or inserted while viewing fluoroscopically, or procedures where the bed is tilted, cannot be done, and so the use of long curtains is limited in procedures that involve endoscopy, such as ERCP, EIS, EST, 201 and ERBD."

Please rephrase for better understandability.

10) page 6, line 202: "This is thought to be one reason why long curtains are used in only 15% of 202 fluoroscopy examinations in the entire hospital ((188+37+32+17)/1834 = 0.149 Table 2)."

Did you count a real usage rate of 15 % in your hospital? Or did you theoretically calculate in which amount of procedures the long curtain could have been used? Please clarify.

11) page 6, line 207: "The population strategy and the high-risk strategy proposed by Rose in 1985..."

I do not feel, the this paragraph really adds value to your paper. This a basic topic of radiation protection, well known to every interventional radiologist. I would propose to omit this topic in your paper.

11) page 7, line 225: Table 2

Please change the first column to left-aligned. This is easier to read.
In the column "Collective PKA" the mean value of all analysed cases is given. Is this correct? Then please add "mean" to the table.

12) page 8

Page numbers are incorrect for page 8-10. Please correct.

13) page 8 (11), line 261: "Looking at fluoroscopy examinations performed in hospitals overall, the reduction rate in the collective equivalent dose for the lens of the physician’s eye was 0.098 with a long curtain and 0.176 with a short curtain, reversing the lens dose reduction effect."

Again, the information is overabbreviated. Please compare "8) page 6, line 195".

Round 2

Reviewer 2 Report

Thank you for revising your manuscript thoroughly. I recommend publication in its current form.